# Occupant Heating Patterns of Low-Temperature Air-to-Air Heat Pumps in Rural Areas during Different Heating Periods

**Xiaoyi Chen** [1,2,3] 🆔, **Ziqiao Li** [1,2,3], **Longkang Dai** [1,2,3], **Wenmao Zeng** [1,2,3] **and Meng Liu** [1,2,3,*]

[1] School of Civil Engineering, Chongqing University, Chongqing 400044, China
[2] Joint International Research Laboratory of Green Building and Built Environment (Ministry of Education), Chongqing University, Chongqing 400044, China
[3] National Centre for International Research of Low-Carbon and Green Buildings, Chongqing University, Chongqing 400044, China
[*] Correspondence: liumeng2033@126.com

**Abstract:** Understanding the actual heating patterns of air-to-air heat pumps (AAHP) in rural areas is crucial for energy planning and clean-heating policy optimization. To explore the influence of outdoor climate change on occupants' heating patterns when using AAHPs in rural areas, the heating season was classified into three periods: the early heating period (EH), the mid heating period (MH), and the late heating period (LH). The investigation was conducted in rural areas of northern China, and indoor environmental parameters were measured from December 2021 to March 2022. Occupants completed household questionnaires about their heating habits before heating and phone interviews at the end of heating. This paper proposes clustering analysis to identify the AAHP heating pattern in rural areas. The results revealed four typical heating patterns of AAHP utilization. Occupant heating behaviors were dominated by the outdoor temperature fluctuation. In particular, during the mid heating period, the heating operation time periods and the heating duration were longer than that of other heating periods. Moreover, the heating patterns in living rooms were different from that in bedrooms. Room occupancy had an impact on household heating demands. These results could provide guidance for energy planning and the development of clean heating policy in the rural area.

**Keywords:** clustering analysis; air-to-air heat pump; heating patterns; heating operation time periods; heating duration





## 1. Introduction

### 1.1. Background

In recent years, severe air pollution occurred in China's northern regions, which has affected people's lives [1]. The pollution showed obvious seasonal characteristics which were closely related to coal-fired heating in the rural areas of northern China [2]. To improve atmospheric quality, the Chinese government put forward the definition of "clean heating" and 'coal-to-electricity' projects as one of the clean heating measures in northern China [3]. Air-source heat pumps (ASHP) are widely used because of their flexible control, high efficiency, and energy-saving potential [4,5]. Moreover, the low-temperature air-to-air heat pump (AAHP) is one of the ASHP that has been widely promoted and the low-temperature air-to-air heat pump is deemed suitable in northern rural China [6]. Current research shows that rural buildings' actual heating demands are intermittent, "part-time, part-space" [4]. Low-temperature AAHP complies with this heating characteristic, which can satisfy occupants' flexible heating demands. In 2017, 2.9 million ASHP units were installed in China [7], and the trend of using ASHP heating is still growing, with the advancement of clean heating, while electricity demand is increasing [8].

Compared with urban buildings, the distributed layouts, building envelope thermal performance, household socioeconomic status, occupant lifestyle, and individual preferences lead to specific demands for China's rural buildings in winter [6]. Moreover, the

daily routine of rural residents also differs from that of urban residents [8]. Regarding the indoor thermal condition in winter seasons, the acceptable temperature range in rural households was lower than the national standard [4]. In addition, the heating approach in rural households was often decentralized, which was different from the centralized heating applied in urban regions [9]. Thus, all the above reasons can potentially impact the heating pattern and occupant heating behavior in rural households (e.g., heating duration, heating period, and heating intensity), which may further influence the power grid load profile in rural areas [10].

### 1.2. Literature Review

Considering the potential determinants of household heating behavior, existing studies have analyzed the behavior of utilizing clean heating in China's northern regions, including the aspects of household background factors, individual subjective factors, and external objective factors.

Rural residents' energy consumption is related to their household structure and economic level. Existing research has examined the household background of rural residents as a factor influencing residents' clean heating utilization. For instance, Yan et al. [11] studied the factors influencing rural residents' choice of clean energy and discovered that per capita income and household permanent population were the most significant factors influencing energy-usage behavior. Furthermore, clean heating behavior is affected by their household size, housing conditions, and expenditure [11]. Households with a single generation may have more difficulty in accepting new energy sources than households living with two generations [11]. Education level, age, and house size are also important factors affecting residents' clean heating usage [12–15]. Rural residents with higher education levels and knowledge about renewable-energy-related policies are more likely to choose clean energy [13]. In addition, when adopting renewable energy, residents consider their economic affordability based on their income level, but their willingness to pay for using renewable energy decreases with the increase in occupant age [13,14]. Li et al. [15] also found that gender, age, and house size are the key factors that influence households' clean heating behavior.

Rural dwellings with dispersed distribution and independent heating control result in greater flexibility in rural heating [9]. Therefore, rural residents' clean heating patterns and behaviors are more vulnerable to individual subjective factors. The previous literature has studied the actual heating demands of rural residents, occupants' control behaviors, and rural residents' satisfaction with the clean heating policy. These previous studies mainly focus on the operation mode of heating equipment, including switching it on and off [4], temperature setting [9], thermal sensation, indoor behavior [16], willingness to pay for clean energy [17], satisfaction with subsidy policies [18], air quality, and the perception of fairness [19]. Previous studies [4,9,16] examined occupants' heating behavior during the heating season. The result indicated that occupants' heating behaviors vary, and their adjustment of heating facilities was based on their perception of thermal sensation. On the other hand, Gong and Liu et al. [17,18] explored residents' willingness to pay for clean heating and evaluated their satisfaction with the subsidy policy, which indicated that a fiscal subsidy is positively associated with the willingness to pay and affects rural residents' satisfaction. Furthermore, Xu et al. [19] found that resident satisfaction with clean heating is significantly influenced by the perceived fairness of the policy implementation process.

Previous studies on promoting clean heating acceptance and understanding how occupant behavior influences heating energy consumption have been conducted from the perspective of external objective factors, such as heating equipment operating parameters, which include fan speed, timer [4,9] supply, and return water temperature, and system power consumption [20]. Liu et al. [21] compared the total annual costs for rural households to adopt different clean heating equipment from the perspective of whether there is a subsidy or not. Of these, building energy efficiencies, average outdoor temperatures, desired indoor temperatures, heating days, and prices of devices and fuels are considered

objective influencing factors, having positive or negative impacts on the clean heating behaviors of rural residents [21]. Moreover, Wang et al. [22], from the perspective of subsidies, fiscal expenditure, and regional heterogeneity, explored energy subsidies' influence on household non-energy expenditure. The results showed that subsidies play a significant role in the clean heating program.

During winter, the outdoor temperature decreases and then increases with the change in heating time from the start to the end of heating. In existing studies related to central heating, to analyze winter heating energy consumption characteristics of buildings, the heating period is classified into three different periods, and each period takes a different heating operation strategy [23,24]. These operation methods include temperature control and the temperature control of the variable flow rate in stages [25], indicating that energy consumption is different for different heating periods and that energy savings can be realized. In addition, some studies divided the heating season into different heating periods and mainly focused on people's thermal comfort characteristics in different heating periods. For instance, Ning et al. [26] conducted a study on students' thermal comfort for different heating periods, and the results showed that the deviations between the mean thermal sensation votes (MTS) and predicted mean votes (PMV) were discrepant in different heating periods. Wang et al. [27] analyzed human thermal comfort and adaptation in an overheated environment in residential buildings, and the research showed that indoor air temperatures in different heating periods are slightly higher than thermal neutral temperatures.

To sum up, in contrast to central heating in urban areas, rural residents may be influenced by a variety of factors and are more likely to control heating equipment or systems based on their demands, such as household background factors [11–15,22], individual subjective factors [4,9,16–19,28], and external objective factors [4,9,20,21]. However, these studies on heating for rural residents ignore the effect of outdoor temperature changes on residents' heating behavior and do not address different heating periods. Furthermore, current studies on heating demand for different heating periods have mainly focused on urban areas rather than rural areas.

Although earlier work has analyzed the influencing factors of households' clean heating behavior (shown in Table 1), these studies did not further divide the heating season and few studies have analyzed characteristics of heating patterns over time in rural areas. Ding et al. [4] found that the heating devices were usually turned on at night and switched off at noon, which indicated that occupants' heating demand was dependent on the outdoor temperature and solar radiation. In China's cold climate zones, the heating season lasts nearly four months (about 120 days), from mid-November to mid-March, with outdoor temperatures ranging from −10 °C to 15 °C. ASHP is a new dispersed load, and with large-scale application, ASHP heating will have an impact on the supply and demand of the grid [29], above all for low-voltage distribution networks where rural residential buildings are connected [30]. Then, the occupant's heating patterns will affect the electricity consumption and regional power load. Therefore, knowledge of ASHP heating patterns and behaviors is important for future power planning, operation, and optimization. Therefore, the whole heating season should be classified into different heating periods based on outdoor temperature and heating time, and the characteristics and differences in rural residents' heating patterns in different heating periods should be analyzed further.

The contribution of this study is to propose a method for a combination of indoor temperature and k-means clustering analysis to identify the heating patterns of AAHP in rural areas of northern China. The remainder of this paper is organized as follows: Section 2 describes the proposed method, which includes a heating season classification, data collection methods, and the AAHP heating pattern identification method. Section 3 presents the results of the AAHP heating pattern in different heating periods in each region. Section 4 discusses the AAHP heating characteristics that are the outcome of this research. Section 5 analyses the limitations of this work and makes recommendations for future research. Section 6 summarizes our study conclusions.

Table 1. Influencing factors of households' clean heating behavior.

| Factor Types | Considering Factors | Method | The Main Objective of the Study |
|---|---|---|---|
| Household background factors | Family size, housing conditions, per capita income, expenditure, and permanent population [11]; Heating area [12]; Age, gender, education level, and income [13–15]. | Questionnaire Logistic regression models | Key factors affecting farmers' choice of clean energy; Acceptance of renewable energy; Clean heating choice. |
| Individual subjective factors | On/off information, mode, temperature setting [4,9]; Thermal sensation and indoor behavior [16]; Occupant control patterns [28]; Satisfaction with clean heating policy, and willingness to pay for clean energy [17], Satisfaction with fiscal subsidies, policy implementation supports, and project effect [18]; Satisfaction with heating level, perceived fairness, air quality, and subsidy amount [19]. | Field investigation Questionnaire Clustering analysis | Actual heating demand patterns; Occupant control behavior; Daily behavior pattern; Acceptance of clean heating; Evaluate rural residents' satisfaction with the clean heating policy; Resident satisfaction with "coal-to-gas" |
| External objective factors | Power, fan speed, and timer [4,9]; Without subsidies and with subsidies, energy prices [21]; Building energy efficiencies, average outdoor temperatures, desired indoor temperatures, heating days, prices of devices and fuels, subsidies, etc [21]; Coal-to-gas subsidies, fiscal expenditure, and regional heterogeneity [22]; Outdoor climate parameters, supply, and return water temperature, and system power consumption [20]; | Simulation platform Artificial neural network Open data | Operation pattern recognition; Economic costs of various clean heating technologies; Occupant control behavior and heating demand. |

## 2. Methodology

### 2.1. Framework

This study consists of four steps: (1) Heating season classification; (2) Data collection and preprocessing; (3) Analyses of the AAHP heating patterns; (4) Discussion of different heating patterns. Firstly, based on the outdoor temperature and heating operation time, we classified winter into three different heating periods. Then, we conducted an on-site survey and monitored the indoor environmental parameters. We proposed a method for identifying AAHP heating operation state by pre-processing the indoor temperature data collected from rural dwellings. Thirdly, a k-means clustering approach was applied to identify the AAHP heating patterns and explore the characteristics of occupant heating patterns' during three different heating periods in winter. Finally, this paper summarized the four typical heating patterns of AAHP in China's northern rural areas and discussed the differences in heating patterns of different heating periods and different room types. The research framework is shown in Figure 1.

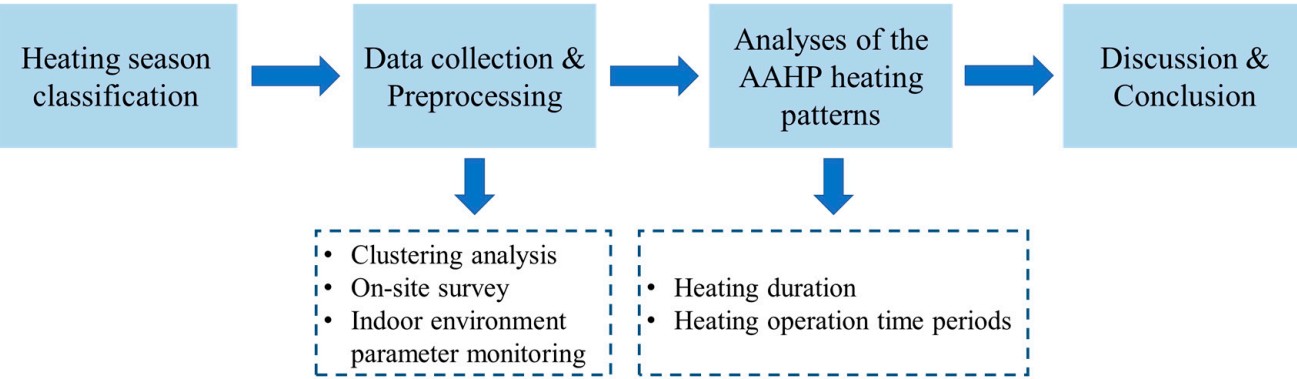

**Figure 1.** The framework of the research.

## 2.2. Heating Season Classification

The annual average daily outdoor temperature in different cities in the northern region of China in winter is shown in Figure 2. (Weather data from DeST 3.0, a building energy simulation software with a Chinese database [31]). The cities are primarily located in a cold climate zone with an average outdoor temperature from −10 to 15 °C, and a heating season lasting nearly four months [32].

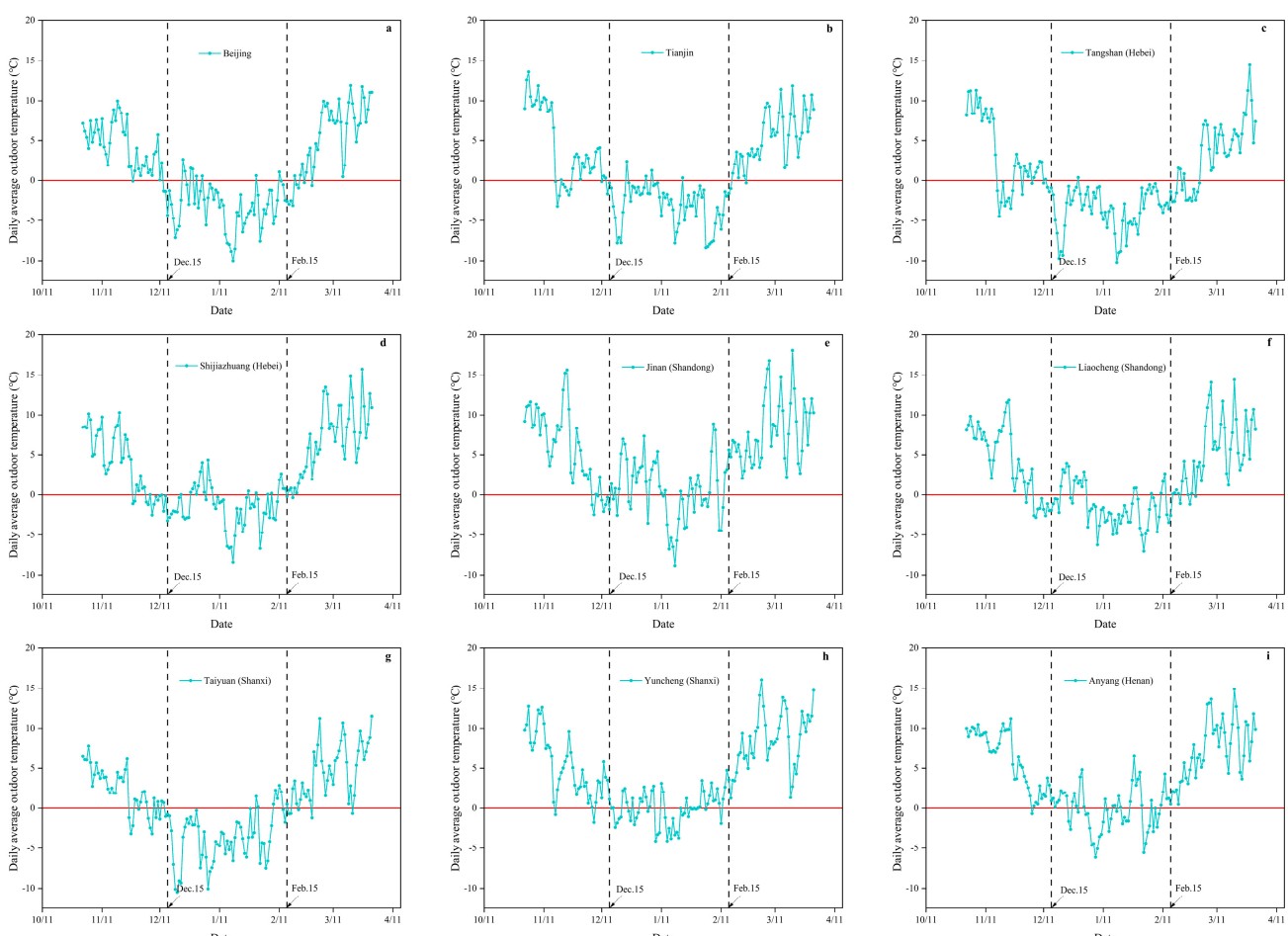

**Figure 2.** Annual average daily outdoor temperature in winter in different cities of the northern region (**a**). Beijing; (**b**). Tianjin; (**c**). Tangshan; (**d**). Shijiazhuang (Hebei); (**e**). Jinnan (Shandong); (**f**). Liaocheng (Shandong); (**g**). Taiyuan (Shanxi); (**h**). Yuncheng (Shanxi); (**i**). Anyang (Henan).

Outdoor temperatures significantly fluctuate in this region during the heating season. From mid-November to mid-December, the outdoor temperature is above 0 °C, which then drops to between −10 °C and 5 °C from mid-December to mid-February. The outdoor temperature commonly begins to increase after mid-February. As shown in Figure 2, there are noticeable temperature turnarounds in mid-December and mid-February in the northern regions.

To identify the occupant heating patterns of AAHP in rural households as the climate changes, based on the outdoor temperature and heating time, the whole heating period was classified into three different heating periods: the early heating period (EH) from November 15 to December 15, the mid heating period (MH) from December 16 to February 15, and the late heating period (LH) from February 16 to March 15 [23,24,33]. The outdoor climate was much colder during the mid heating period.

*2.3. Data Collection and Preprocessing*

2.3.1. On-Site Survey and Monitor

The housing type, household structure, heating pattern, and AAHP heating utilization are taken into consideration in order to investigate the heating pattern of AAHP in northern rural areas. The researchers in this study conducted face-to-face questionnaire surveys and field investigations in three northern cities, namely Dingzhou, Hebei Province; Jining, Shandong Province; and Yuncheng, Shanxi Province, during the heating season (from December 2021 to March 2022). Thirty households were surveyed regarding building information, family information, and AAHP heating utilization. Moreover, a post-evaluation was conducted after the heating season ended to collect information about occupants' AAHP heating habits via telephone interviews. A total of 28 households were interviewed. The results of the on-site survey are shown in Table 2.

**Table 2.** Building information, family information, and AAHP heating usage in three different regions in northern China.

| | Region | Hebei | Shandong | Shanxi |
|---|---|---|---|---|
| Building information | Building floors | Two or three story | Single-story | Single-story |
| | Room height/m | 3 m | 3.5 m | 3.8 m |
| | Permanent rooms/room | 1–2 | 2–3 | 2–3 |
| Family information | Permanent population/person | 2–3 | 3–4 | 3–4 |
| | Maximum population/person | 3–4 | 4–5 | 4–5 |
| | Number of households surveyed/household | 10 | 10 | 10 |
| | Number of households interviewed by phone/household | 9 | 10 | 9 |
| AAHP heating usage | AAHP capacity | 1.5 P | 2 P | 2 P |
| | Installed number | 1 | 1 | 1 |
| | Room with AAHP installed | Bedroom | Living room | Living room |
| | Heating area/m$^2$ | 10–15 | 30–35 | 20–25 |
| | Heating electricity price CNY/kWh | 0.55 (08:00–20:00) 0.3 (20:00–08:00) | 0.3 | <2600 kWh, 0.18 >2600 kWh, 0.48 |
| | Average heating energy consumption/kWh per household | 2200 | 750 | 1200 |

(Note: CNY indicates Chinese yuan; 'P' indicates the input electrical power of the AAHP, 1 P = 735 W).

According to our investigation, single-story and two- or three-story buildings are the most common in rural areas in the northern. As shown in Table 2, 10 rural households surveyed in Hebei Province have two- or three-story buildings, and the houses in Shandong and Shanxi provinces have a single story. In Hebei province, the occupants choose to live on the upper floors (second or third floor), facing to the south, to receive more solar radiant

heat in the winter. The bedroom is the main place for occupants' daily activity and resting during the heating seasons, resulting in most AAHP equipment being installed in bedrooms. In Shandong and Shanxi provinces, the building are all single-story structures and the living room is generally located in the center of the entire rural building, with 1–2 bedrooms connected to the living room. Most daily activities take place in the living room in both Shandong and Shanxi provinces, and the AAHP heating equipment is all installed in the living room. The three provinces of Hebei, Shandong, and Shanxi are all located in northern China, which is a cold climate zone and there is a high demand for heating in the winter. The building types and household structures have common points. The price of heating electricity differs by region. Hebei charges 0.55 CNY/kWh for peak electricity and 0.3 CNY/kWh for valley electricity; Shandong charges 0.3 CNY/kWh; Shanxi charges 0.18 CNY/kWh for up to 2600 kWh and 0.48 CNY/kWh for over that amount. The average household heating energy consumption in Hebei, Shandong, and Shanxi is 2200 kWh, 750 kWh, and 1200 kWh, respectively, according to the research.

Although AAHP is the main heating device for space heating in each region, residents in Shandong and Shanxi provinces also use air conditioners, electric blankets, and oil-filled electric heaters for local heating, as shown in Figure 3.

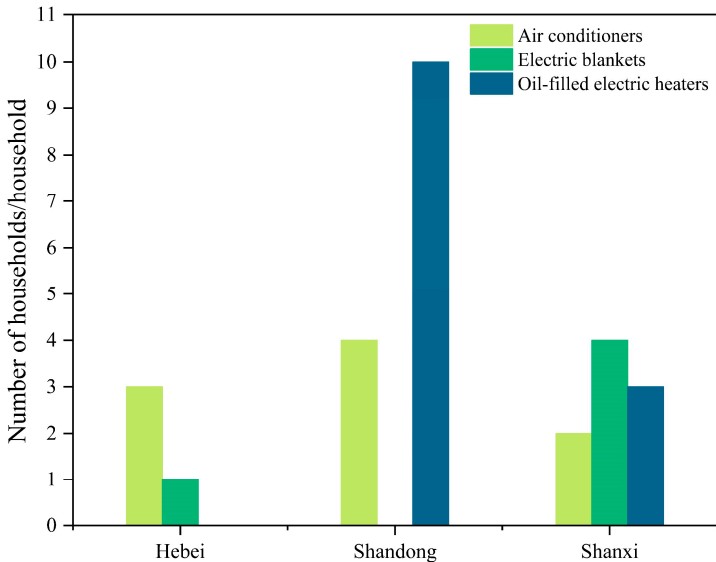

**Figure 3.** The use of auxiliary heating equipment.

We conducted field measurements on AAHP heating usage in three regions in Hebei, Shandong, and Shanxi provinces, China. During winter, indoor environmental parameters, such as humidity and temperature, were measured. A temperature/humidity logger was mounted on the wall with the AAHP installed in each investigated room, which was positioned to avoid direct airflow from the AAHP outlet. The monitoring investigation was from 1 December 2021, to 31 March 2022, and the indoor environmental condition was recorded every hour. Detailed descriptions of the monitoring devices are shown in Table 3.

**Table 3.** Indoor environment parameters monitoring device.

| Parameters | Instruments | Range | Resolution and Error Range |
|---|---|---|---|
| Humidity and Temperature | Humidity and temperature logger (ONSET HOBO UX100-003, America) | −20 °C–70 °C 15–95% | Accuracy ± 0.21 °C Accuracy ± 3.5% |

### 2.3.2. Analysis of AAHP Operational Status

Building upon the common situation that the indoor temperature change would depend on the operation status of AAHP heating in these investigated rooms, we used the indoor-temperature variation trend to reflect the operation of AAHP heating equipment.

A database of the AAHP's operational state at every recording of the day can be obtained from Equation (1) after pre-processing the room indoor temperature data.

$$\Delta T_i = T_i - T_{i-1} \tag{1}$$

where $T_i$ is the room temperature at the (i) th hour, °C; $T_{i-1}$ is the room temperature at the $(i-1)$ th hour, °C, $i \in [0, 23]$; $\Delta T_i$ is the difference in indoor temperature between the (i) th and $(i-1)$ th hours, °C.

$$\Delta T_{i(i-o)} = T_i - T_{iout} \tag{2}$$

To avoid misjudging the AAHP heating operating status caused by the fluctuating indoor temperature resulting from solar radiation and outdoor temperature changes, the difference between indoor and outdoor temperatures was introduced as a limiting condition, as presented in Equation (2), where, $T_{i(i-o)}$ is the difference in indoor temperature and outdoor temperature in the (i) th hours, °C; $T_{iout}$ is the outdoor temperature in the (i) th hours, °C.

After the AAHP was turned on, the effective indoor temperature, temperature fluctuation value, and indoor-outdoor temperature difference value were used as important indicators to judge the operation status. Meanwhile, to reduce the influence of short-term temperature fluctuations on the judgment of the AAHP operation status over a period, the study proposed Equation (3) to regulate the AAHP operation identification criteria.

$$F(i) = \begin{cases} 1, & i \in A, A = \left\{ i \ \middle| \ T_i > T_0, \Delta T_{i(i-o)} > T_1 \right\} \\ & \cup \{i \mid \Delta T_i > T_2\} \\ & \cup \{i \mid \Delta T_{i-1} > T_2, \Delta T_i > T_3\} \\ & \cup \{i \mid \Delta T_{i-2} > T_2, \Delta T_i > T_3\} \\ & \cup \{i \mid \Delta T_i > T_2, \Delta T_{i+1} > T_3\} \\ & \cup \{i \mid \Delta T_{i-1} > T_2, \Delta T_{i+1} > T_3\} \\ 0, & i \in A, A = \complement_Q A \end{cases} \tag{3}$$

where $F(i)$ is the operation state in the (i) th hour; $Q$ is the set of rational numbers, and $\complement_Q A$ denotes the complement of set $A$. If the AAHP is on, the value is 1; if the AAHP is off, the value is 0.

According to the design standard of energy-efficient rural housing, the heating indoor design temperature is 14–16 °C in rural areas [34], and we selected $T_0 = 14$ °C as the effective indoor temperature to judge whether the AAHP was turned on.

To avoid solar radiation and outdoor temperature affecting the indoor environment. We referred to the value of the 5 °C temperature difference between the non-heated room and the adjacent heated room [35]. Then, in this study, we chose $T_1 = 5$ °C as the temperature difference between the indoor and outdoor temperature to judge the AAHP's operating status. Additionally, $T_2 = 0.5$ °C, $T_3 = 0$ °C was chosen as the value to judge the temperature fluctuation.

Yan et al. [36] applied cluster analysis to identify the temperature-setting patterns for room air conditions. An et al. [32] used cluster analysis to identify air condition use patterns in residential buildings. Thus, in this study, to identify AAHP heating patterns, the k-means clustering method was chosen to categorize the daily AAHP heating patterns by the on/off operation status identified by Equations (1) and (2). Clustering analysis is a process of partitioning a set of observations into subsets in a way that objects belonging to the same cluster have a high similarity, while objects belonging to different clusters have a low similarity [37]. This is achieved with the use of various cluster algorithms, such as k-means and fuzzy clustering [32].

The k-means clustering method groups a dataset of N input vectors to C clusters using an iterative procedure. Initially, the weights of the C clusters are determined, and a random selection of the N input vectors is made for the cluster centroids. The estimated centroids are then used to classify objects into clusters through Euclidean distances, expressed by:

$$d(x,y) = \sqrt{(x_1 - y_1)^2 + (x_2 - y_2)^2 + \ldots + (x_n - y_n)^2} \tag{4}$$

where $x = (x_1, x_2, \ldots, x_n), y = (y_1, y_2, \ldots, y_n)$ are two objects in a Euclidean n-space.

Next, the Euclidean distances of each object of the centroid are recalculated in such a way that each object of the centroid is the average of the object of the load patterns within the cluster. The procedure is repeated until the stabilization of the cluster centroids is achieved.

As shown in Table 4, we proposed two indicators to analyze the AAHP heating pattern characteristic of residents in rural areas of northern China: heating duration and heating operation time periods.

**Table 4.** AAHP heating pattern characteristic indicators.

| Indicators | Definition |
| --- | --- |
| Heating duration | Average daily number of AAHP-on hours |
| Heating operation time periods | AAHP daily operation time periods |

## 3. Results

### 3.1. Clustering Number Selection for the Heating Pattern of AAHPs

In this study, we selected the best cluster number as the input for k-means clustering. Silhouette score (SC) is a common metric index for selecting the optimal cluster number [36]. A cluster number with a higher SC was always preferred. We used k-means clustering with different cluster numbers and chose the cluster number with the highest SC as the final selected cluster number to identify the heating patterns [38]. A small number of clusters cannot help us in identifying meaningful typical and diverse patterns, while a large number of clusters would be difficult to interpret [39]. Therefore, the optimal cluster number was selected to be between 2 and 9.

Clustering analysis was applied to identify typical AAHP heating patterns during daily life to represent diverse heating behavior characteristics. The attributions used are the AAHP on/off states from 0:00 to 23:00.

Figure 4 shows the optimal cluster number of heating patterns of AAHP heating in the bedroom in Hebei rural areas, including the three heating periods of early heating, mid heating, and late heating. The optimal cluster number for each of the three different heating periods is 6, 2, and 2.

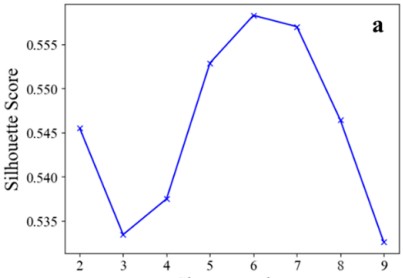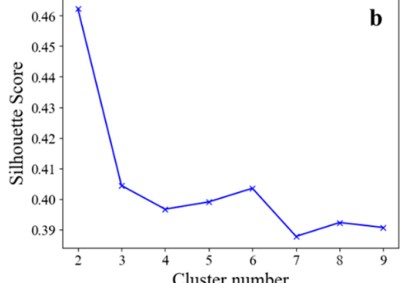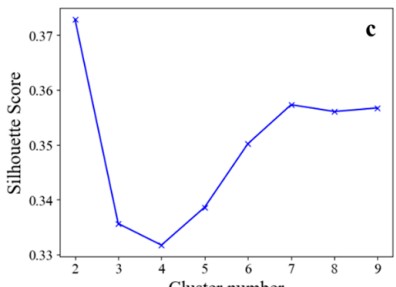

**Figure 4.** Results of the optimal cluster number selection in Hebei's bedrooms ((**a**). Early heating period; (**b**). mid heating period; (**c**). late heating period).

As shown in Figure 5, the optimal number of a cluster for AAHP heating patterns in rural residential living rooms in Shandong Province shows that the optimal cluster number of the three different heating periods is 8, 2, and 2, respectively.

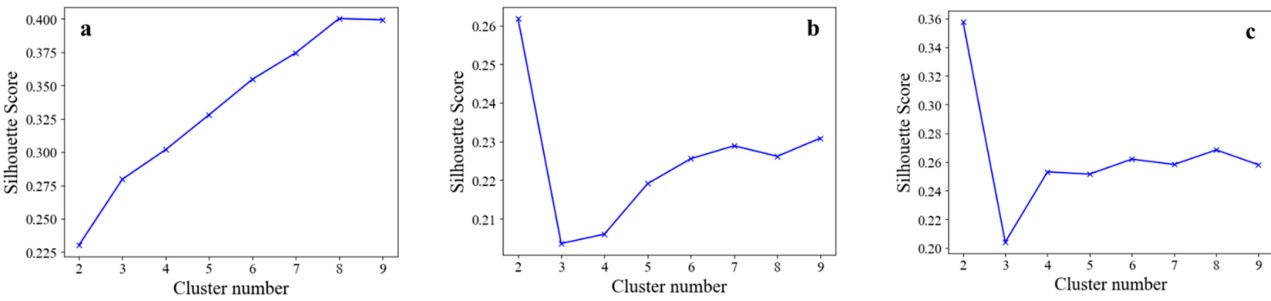

**Figure 5.** Results of the optimal cluster number selection in Shandong's living rooms ((**a**). Early heating period; (**b**). mid heating period; (**c**). late heating period).

As shown in Figure 6, the optimal number of a cluster for AAHP heating patterns in rural residential living rooms in Shanxi Province shows that the optimal cluster number of the three different heating periods is 2. The optimal number of clusters of heating patterns in the three different heating periods is relatively close.

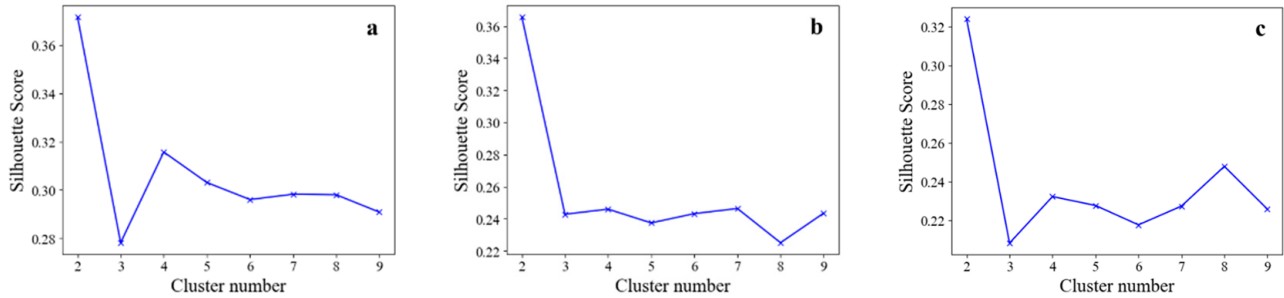

**Figure 6.** Results of the optimal cluster number selection in Shanxi's living rooms ((**a**). Early heating period; (**b**). mid heating period; (**c**). late heating period).

## 3.2. Analyses of AAHP Heating Patterns

### 3.2.1. AAHP Heating Pattern in Rural Areas of Hebei Province

Figure 7 indicates the frequency with which various heating patterns were used during this heating period. We calculated the types and proportions of heating patterns in rural areas of Hebei province based on the clustering results. It can be found that during the early heating period (Figure 7a), the heating pattern of Cluster 1 had the highest proportion, accounting for 50% of all patterns. This means that of the six heating patterns in the early heating period, the most popular heating pattern is Cluster 1. When entering the mid heating period, there are only two heating patterns (53% for Cluster 1 and 47% for Cluster 2). In the late heating period, Cluster 1 accounted for 40%, and Cluster 2 for around 60%. The change in heating patterns reflected the change in rural residents' heating demand during the different heating periods.

We analyzed the percentage of different heating patterns for three heating periods in Hebei's bedrooms and show the ten typical AAHP heating pattern schedules in Figure 8.

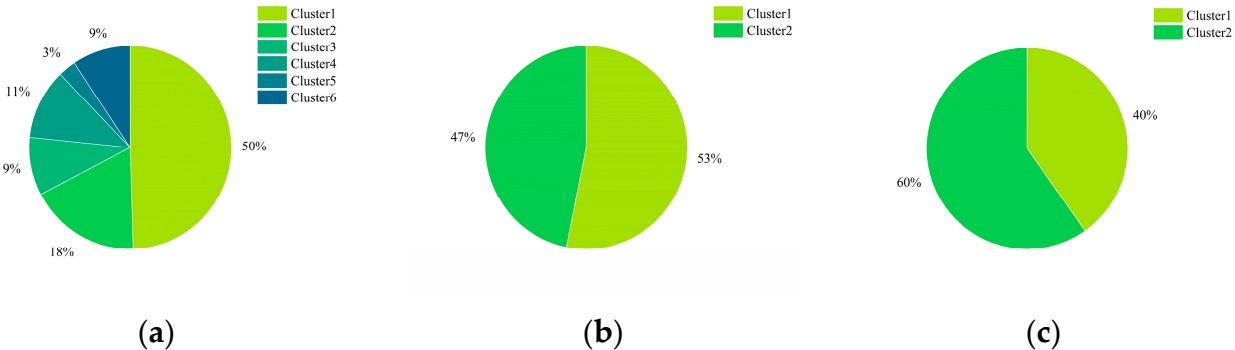

**Figure 7.** The accounted percentage of different heating patterns in Hebei's bedrooms ((**a**). Early heating period; (**b**). mid heating period; (**c**). late heating period).

For the bedroom, during the early heating period, Cluster 1 is the most common heating pattern of the AAHP (accounting for 50%), representing a continuous heating pattern of turning on the AAHP all day. There is a total heating duration of 24 h in Cluster 1. However, the heating pattern in Cluster 2 (18%) represents no heating demand, and the AAHP is always off. Cluster 3 represents the "before dawn-off user" and Cluster 4 the "morning and afternoon-off user". These two heating patterns accounted for a total usage of 9% and 11%. Cluster 5 and Cluster 6 represent the "daytime-on user", the heating duration is 12 h and 7 h, respectively.

Cluster 1 accounts for 53% of the mid heating period and represents an intermittent heating pattern with the AAHP turned on at lunchtime and before sleep. As the weather gets colder and enters the mid heating period, most occupants choose to turn on the AAHP all day. Cluster 2 (which accounts for 47% of the total) represents the "always-on user", which indicates that some occupants spent more time at home and prefer the AAHP to be on all day in this heating period.

There are also two main heating patterns when entering the late heating period. Cluster 1 accounts for 40%, where occupants turn on the AAHP for heating at lunchtime. However, there are still 60% of occupants who use the AAHP 24 h a day for heating during the late heating period.

When an AAHP was installed in Hebei province's bedrooms, residents turned on the AAHP the whole night and preferred using a continuous heating pattern to keep the bedroom environment comfortable. Therefore, the location of the AAHP installation will influence heating patterns.

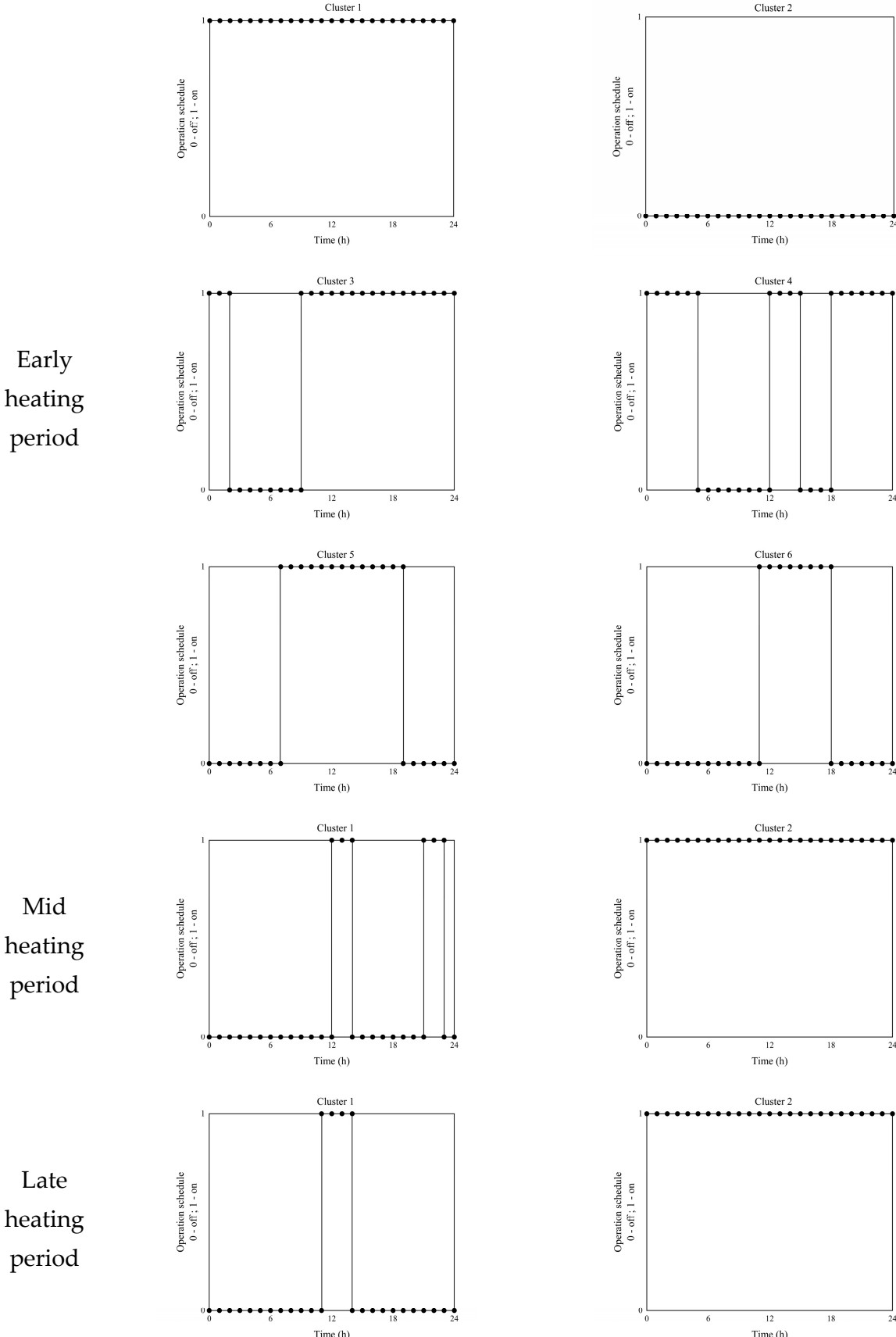

**Figure 8.** AAHP heating pattern schedules for different heating periods in Hebei's bedrooms.

### 3.2.2. AAHP Heating Pattern in Rural Areas of Shandong Province

We calculated the types and proportions of heating patterns in rural areas of Shandong province based on the clustering results. The proportion of each heating pattern in different heating periods are shown in Figure 9. There are eight heating patterns in the early heating period but only two in the mid and late heating period. The proportion of heating patterns in different heating periods differs significantly as well.

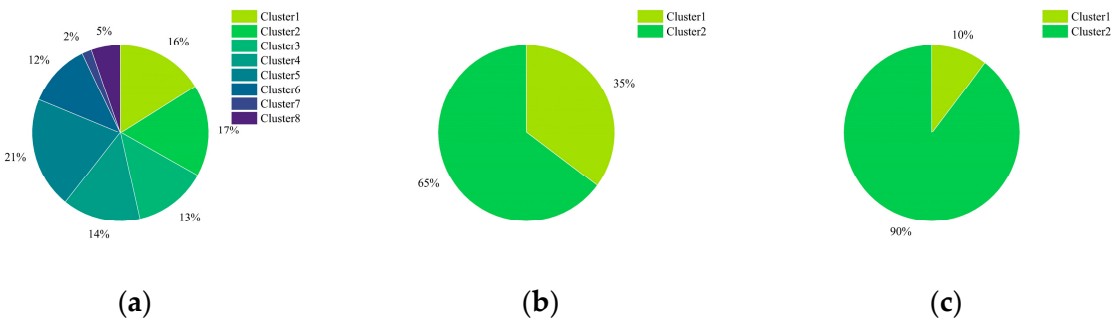

**Figure 9.** The accounted percentage of different heating patterns in Shandong's living rooms ((**a**). Early heating period; (**b**). mid heating period; (**c**). late heating period).

For the living room in Shandong province, as shown in Figure 9, we analyzed the percentage of different heating patterns for three heating periods and summarize the 12 typical AAHP heating pattern schedules in Figure 10.

There are eight heating pattern schedules for living rooms in Shandong Province during the early heating period. In terms of heating operation time periods, these heating patterns are classified into two types: one operates only during the day and another operates not only during the day but also at night. Cluster 1 (16%), Cluster 2 (17%), Cluster 4 (14%), and Cluster 6 (12%) turned on the AAHP at lunchtime or in the afternoon with a heating duration of 3–4 h. Cluster 3 (25%) and Cluster 8 (19%) represents the "noon and evening-on user", and they typical have an intermittent heating pattern. Occupants will turn on the AAHP at lunch and before going to bed, and the heating duration will be nearly 10 h. Cluster 5 represents the "always-off user" and accounted for a total usage of 21%. Cluster 7 primarily operated the AAHP in the evening, before dawn, and in the morning, accounting for about 2%; the heating duration for this Cluster is 14 h.

During the mid heating period, there are two intermittent heating patterns for living rooms in Shandong province. Clusters 1 and 2 accounted for 35% and 65%, respectively. Cluster 1 primarily uses AAHP to heat from noon until bedtime, after which the AAHP heating equipment is turned off, accounting for 13 h. Cluster 2 uses the AAHP for heating at lunchtime and the heating duration is 4 h.

In the late heating period, there are two heating patterns. Cluster 2 only turned on the AAHP during the day from noon to afternoon, while Cluster 1 indicates that occupants not only use the AAHP during the day but also at night after sleep. Cluster 1 (about 10%) represents the higher heating demands, while Cluster 2 (about 90%) represents intermittent heating for about 4 h from 12:00 to 16:00.

In the rural areas of Shandong Province surveyed, the residents mainly work as farmers who commonly stay home for most of their daily time during winter. Although outdoor temperatures might be higher in the afternoon, outdoor temperatures were still too cold to make residents shut down heating devices. Most residents have the habit of having a lunch break and, thus, turn on the AAHP in the period 12:00–15:00 to keep the room warm. Another reason is that rural families might have elderly people and children who will spend most of their time indoors in winter. In addition, room occupancy may also affect the AAHP heating operation time period.

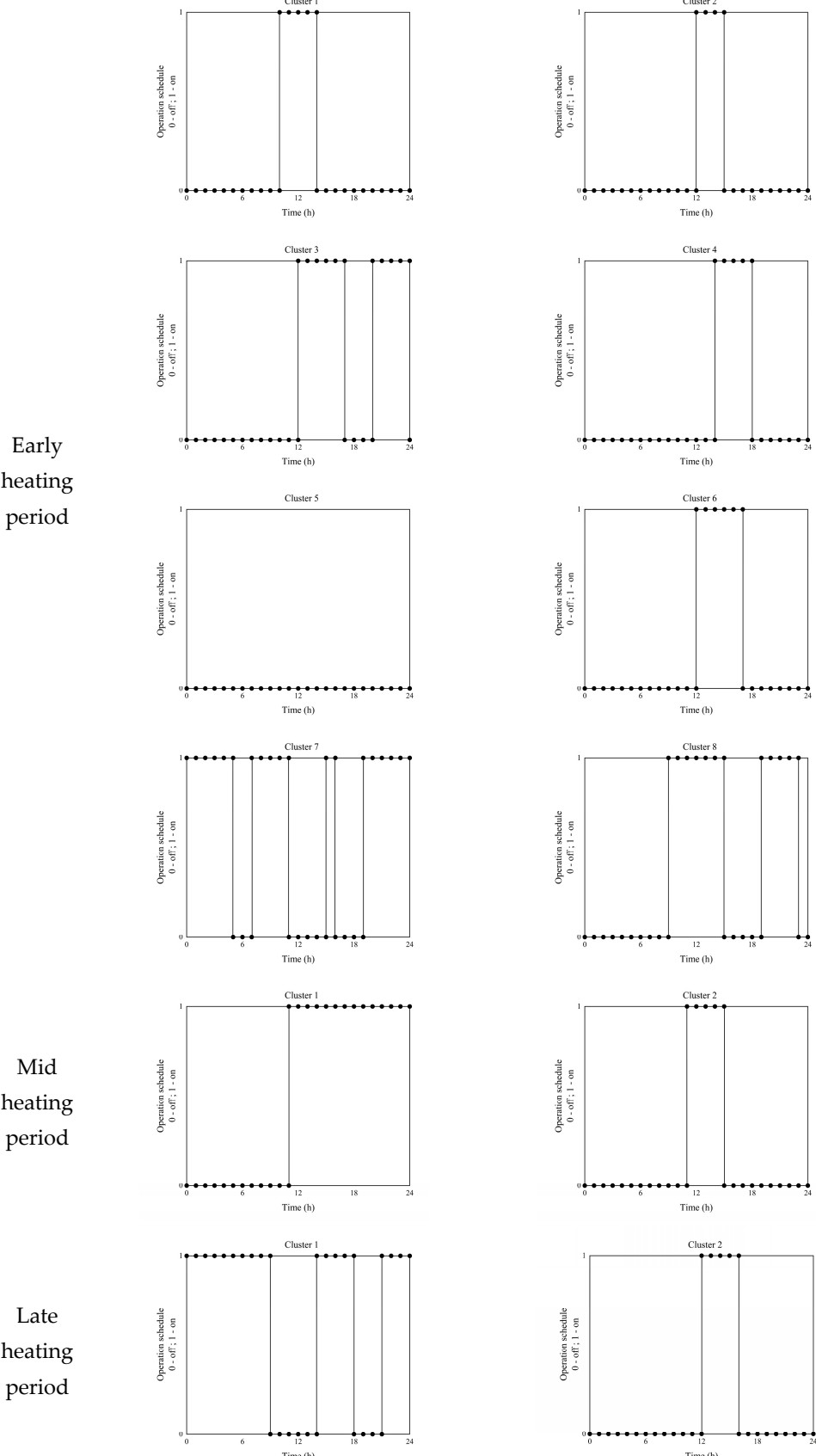

**Figure 10.** AAHP heating pattern schedules for different heating periods in Shandong's living rooms.

### 3.2.3. AAHP Heating Pattern in Rural Areas of Shanxi Province

We calculated the types and proportions of heating patterns in rural areas of Shanxi province based on the clustering results. The proportion of each heating pattern in different heating periods is shown in Figure 11. There are two heating patterns in different heating periods. During the early heating period, Cluster 1 was around 34%, and Cluster 2 has the largest percentage, accounting for 66%. In the mid heating period, Cluster 2 accounted for around 84%, while Cluster 1 accounted the smallest proportion, which was only 16%. During the late heating period, the percentage of Cluster 1 and Cluster 2 was 26% and 74%, respectively.

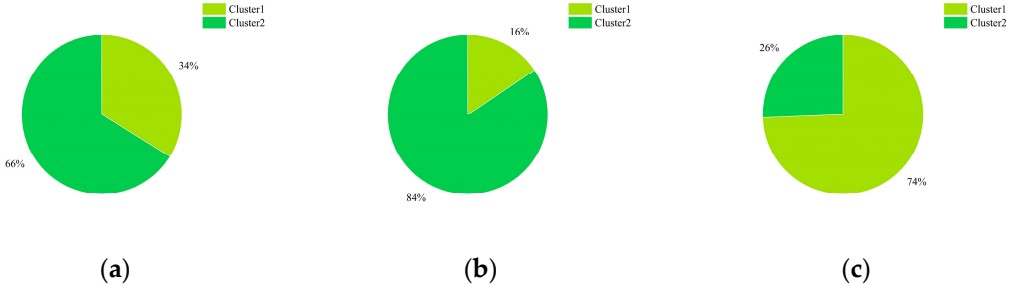

(**a**)         (**b**)         (**c**)

**Figure 11.** The accounted percentage of different heating patterns in Shanxi's living rooms ((**a**). Early heating period; (**b**). mid heating period; (**c**). late heating period).

For the living rooms in Shanxi province, as shown in Figure 11, we analyzed the percentage of different heating patterns for three heating periods and summarize the six typical AAHP heating pattern schedules in Figure 12.

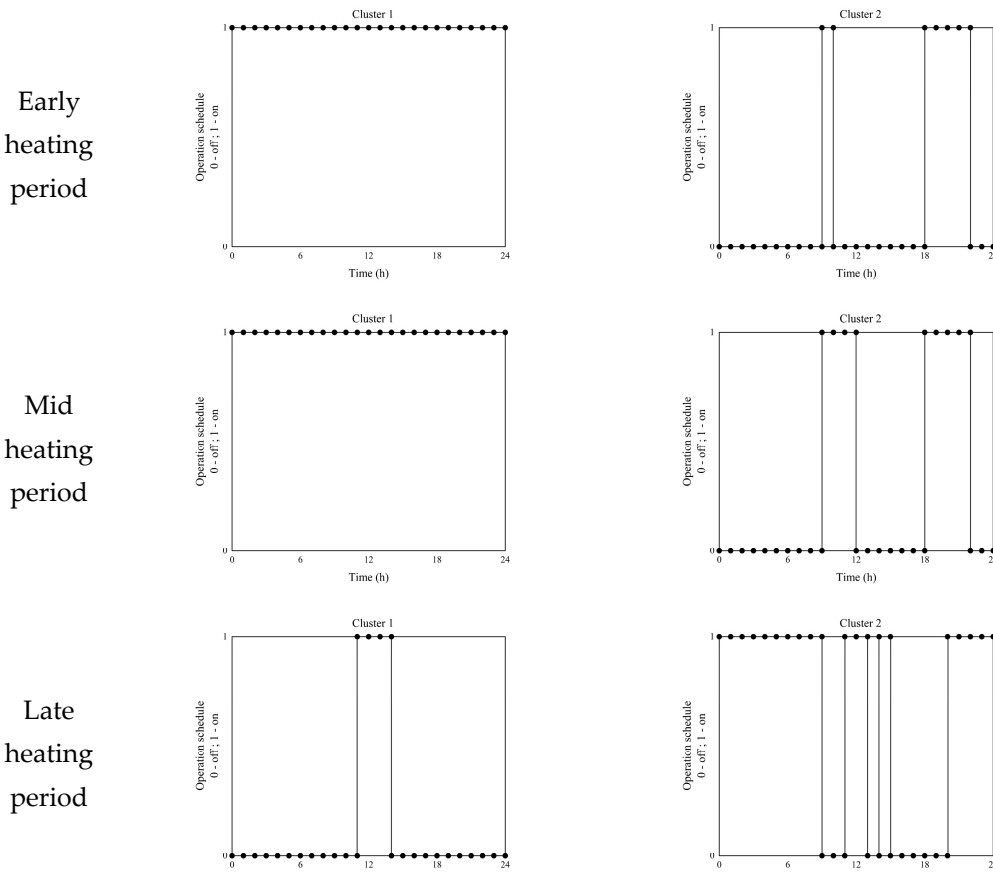

**Figure 12.** AAHP heating pattern schedules for different heating periods in Shanxi's living rooms.

During the early heating period, there are two heating patterns of AAHP heating in the living rooms in Shanxi province. Cluster 1 has a proportion of about 34%, which represents the "always-on user". Cluster 2 accounted for 66% of the early heating period. Overall, more than half of the occupants used the AAHP for intermittent heating in the morning and evening until going to bed. The heating operation time periods were from 9:00 to 10:00 and from 18:00 to 22:00, with a heating duration of 5 h.

There are two heating patterns during the mid heating periods. Cluster 1 accounted for about 16%, with continuous heating all day; in Cluster 2 (accounting for about 84%), the heating operation time periods were from 9:00 to 10:00 and from 18:00 to 22:00, and the heating duration was 7 h a day.

There are two heating patterns in the late heating period. Cluster 1 (accounting for 74%) mainly ran around noon; the heating operation time periods were from 11:00 to 14:00, with the heating duration being 3 h. Cluster 2 (accounting for 26%) turned on the AAHP at noon and the whole night and the heating operation time periods were 12:00 to 14:00, 15:00 to 16:00, and 20:00 to 10:00 the next day, with the heating duration being 15 h.

In the rural areas of Shanxi province, during the different heating periods, some residents will turn on the AAHP for heating from 18:00 to 24:00, which indicates that a large number of residents will heat their homes before going to bed in the evening. At this time of day, rural households have the highest occupancy, and they prefer to use the AAHP to heat their living rooms and turn them off before going to bed.

## 4. Discussion

### 4.1. Typical AAHP Heating Patterns in Rural Areas

The k-means clustering analysis can not only identify the typical heating patterns of different heating periods but also assists us in understanding the characteristics of occupant heating behaviors, such as heating duration and heating operation time periods, which can reflect more detailed information and increase the knowledge about real-world AAHP heating demands in rural areas.

According to the findings of the analysis, AAHP heating in rural areas during the heating season can be classified into two main heating patterns: intermittent heating and continuous heating. Furthermore, intermittent heating can be divided into two categories: "Intermittent heating for multiple periods" and "Intermittent heating for a single period". The former indicates that the AAHP will be turned on/off multiple times during the day, whereas the latter indicates that they will only be turned on/off once.

When the AAHP is used for continuous heating, some occupants would heat their rooms for the whole day, while others would turn off their AAHP for some of the time periods, but the daily cumulative heating time duration was around 14–16 h. According to these characteristics, we defined these two continuous heating patterns as "Continuous heating for 24 h" and "Continuous heating for multiple periods". The typical heating pattern examples are shown in Table 5.

**Table 5.** Typical heating patterns of the AAHP for heating in rural areas.

| Typical Heating Patterns | Examples | | Heating Pattern Characteristics | |
| --- | --- | --- | --- | --- |
| | Heating Operation Time Periods | Heating Duration | | |
| Intermittent heating for multiple periods | 12:00–17:00 and 20:00–23:00 | 8 h | Dispersion | Medium |
| Intermittent heating for a single period | 9:00–14:00 | 5 h | Concentration | Low |
| Continuous heating for 24 h | 0:00–23:00 | 24 h | Continuous | High |
| Continuous heating for multiple periods | 7:00–11:00 and 15:00–16:00 and 19:00–5:00 (next day) | 14–16 h | Dispersion | High |

As shown in Table 5, there are four typical heating patterns in rural areas of northern China. In terms of heating operation time periods and heating duration, we describe the characteristics of various heating patterns, which could reflect the occupant's AAHP heating intensity.

In the case of the intermittent heating pattern, "Intermittent heating for multiple periods", this heating pattern consists of multiple heating operation time periods and the characteristics were dispersion, with the household AAHP use intensity at a medium level. For the "Intermittent heating for a single period" pattern, this heating pattern consists of a single heating operation time period and the characteristics were concentration, with the household AAHP use intensity at a low level.

In the case of the continuous heating pattern, "Continuous heating for 24 h", this heating pattern represents the "always-on user" and the characteristics were continuous, with the household AAHP use intensity at a high level. For the "Continuous heating for multiple periods" heating pattern, this heating pattern consists of multiple heating operation time periods, where occupants will turn on/off the AAHP many times and the characteristics were dispersion, with the household AAHP use intensity at a high level.

AAHP heating equipment was completely under the control of rural residents, who could make immediate adjustments if they were dissatisfied with their thermal comfort, demonstrating the flexible control characteristic of AAHP heating devices. As a result, for rural residents who use AAHP heating, some intermittent and continuous heating patterns have emerged. These heating patterns are consistent with rural areas' "part-time, part-space" heating characteristics.

*4.2. AAHP Heating Pattern Characteristics in Different Heating Periods*

As the results demonstrated, the heating operation time periods and the heating duration in the mid heating period was higher than that of the other two heating periods. When the outdoor temperature in Figure 2 is compared to the AAHP heating pattern operation characteristics in different heating periods in Figures 8, 10 and 12, it can be seen that the heating pattern of AAHP operation could be affected by the outdoor air temperature change. The data analysis results show that the optimal number of clusters for the heating pattern of AAHP in different heating periods is shown in Figures 4–6. During the early heating period, the heating pattern cluster number is higher than the mid and late heating periods, which indicated that the occupants' heating demand will adjust according to the outdoor temperature change and their thermal comfort. When entering the mid heating period, occupants are more likely to use a fixed heating pattern for heating rather than frequently adjusting their heating equipment. Meanwhile, heating operation time periods and heating duration will increase compared to the other heating periods.

In the rural area of Hebei province, the "always-on" heating pattern accounted for nearly half of the total heating patterns in each heating period. However, the difference is that in the early heating period, there was an "always-off" heating pattern, which is approximately 18% of the total heating patterns. Some occupants did not stay at home every day (e.g., for job-related reasons) or they did not feel cold (e.g., because the outdoor temperature was high and or they wore more clothes) so they did not have the same heating demands. Furthermore, some occupants switch the AAHP on/off more frequently owing to the outdoor temperature change. Therefore, there are six different types of heating patterns in the early heating period. The heating patterns will change as the outdoor temperature drops and enters the mid heating period, and as the outdoor temperature gradually rises and enters the late heating period. Each heating period had only two types of heating patterns (e.g., continuous heating for 24 h and intermittent heating for multiple periods). Heating patterns, however, differ between the mid and late heating periods. During the mid heating period, occupants will use the AAHP for heating not only during lunchtime but also at night before sleep. However, in the late heating period, they only use the AAHP at lunchtime. Except for those in the continuous heating patterns, the mid heating period's heating operation time periods are higher than the late heating period, indicating a

higher heating demand in the mid heating period. In Hebei province, most elderly people always stay at home, and the AAHP is installed in the bedroom, which is also the main space for people's daily activities, so the AAHP will always be turned on to keep the room comfortable.

In the rural area of Shandong province, approximately 90–98% of houses had an intermittent heating pattern in three heating periods. Figure 10 shows the common characteristic of heating patterns for the three different heating periods, which is that the occupants prefer to turn off the AAHP after they leave the living room for sleep. During the early heating period, some people did not use the AAHP for heating and the equipment was "always off". In this period, the outdoor temperature is relatively higher than in other periods, and some people did not use the AAHP to save on heating costs. The early heating period's heating pattern has eight different types, which is more than the other two heating periods, and this reflected the heating demands was changed. Compared with the intermittent heating pattern in the remaining two heating periods, this heating pattern has the characteristics of concentration, which reflects the residents' demand and tendency for continuous heating in the mid heating period. During the mid heating period, occupants will turn on the AAHP from noon until they go to bed in the evening. However, half of the heating patterns were ones which used the AAHP in the afternoon, because during this time, people usually stay at home. During the late heating period, approximately 10% of the heating patterns were "continuous heating for multiple periods." Because the outdoor temperature is still low during this heating period, some households with children may choose to keep their heating demands. However, as the outdoor temperature rises, people will reduce the AAHP operation time, only using the AAHP when the majority of residents are at home, believing that it is worthwhile to switch on the AAHP for heating. They usually wear more clothes at home and are constantly moving indoors and outdoors based on their daily routine. They prefer to use an intermittent heating pattern to reduce heating operation costs.

In Shanxi province's rural areas, the AAHP installed in the living room has two types of heating patterns in each heating period. The heating operation time periods and heating duration of the intermittent heating pattern were greater in the mid heating period, as shown in Figure 12. Regarding the heating duration, especially the intermittent heating pattern in each heating period, the average daily operation hours were higher in the mid heating period. During the late heating period, the continuous heating pattern changed to the continuous heating for multiple periods pattern, which also indicates that the heating demand changed. Based on the change in outdoor temperature, occupants will adjust their heating pattern and heating demand. Approximately 16–34% of the heating patterns had higher heating demands (including the two heating patterns "Continuous heating for 24 h" and "Continuous heating for multiple periods"). Because these families have children, they must use the AAHP to heat the room for children to be comfortable. Most people use intermittent heating patterns because they believe that "electricity heating" is more expensive than coal heating, and they prefer to heat the room with an intermittent heating pattern, which could reduce heating costs. Our survey also found that residents generally stated that they would ignore the impact of economic issues when the weather cools down to ensure comfort. However, when the outdoor temperature rises, they will still adjust the AAHP operation to achieve a balance between comfort and economy. Therefore, the economic level of rural residents and their knowledge of clean heating could be potential factors in their heating behaviors.

### 4.3. The Heating Patterns of AAHP for Different Room Types

There are differences in the rooms where AAHP are installed in each region, which can also have an impact on occupant usage. The analysis of the AAHP heating pattern in bedrooms revealed that more than half of the heating patterns are continuous heating for 24 h. If the AAHP is installed in the living room, most residents prefer to use the intermittent heating pattern and turn off the AAHP after going to bed. Therefore, heating

patterns differ when AAHP is used due to differences in AAHP installation locations and residents' living habits.

In addition, based on the responses of residents, those who do not need to work spend more than half of their daytime in bedrooms during the winter, and the rest of the rooms are only used for short periods each day. When they were satisfied with their thermal comfort or left the bedroom, they tended to turn off the AAHP, primarily to save on heating operation costs.

As shown in Figure 9, the optimal number of clusters for the different heating patterns of AAHP heating in rural areas of Shandong province shows that the number of heating pattern types in the early heating period is significantly higher than in the other two heating periods. The heating pattern of different heating periods is the intermittent heating pattern, which is mainly used during the day and before going to bed. During the mid heating period, the number of heating pattern categories in the mid heating period is the lowest. The outdoor temperature in the mid heating period is the lowest throughout the whole heating season and the heating demand is the highest. Therefore, it is indicated that the occupants who use AAHP heating will adjust the heating pattern according to the changes in the outdoor temperature.

Shown in Figure 11 is the optimal clustering numbers of the operation patterns of the AAHP in different heating periods in rural areas of Shanxi. In this Figure, it is shown that the number of heating pattern categories in the three heating periods has little difference. The heating pattern of living rooms in rural areas of Shanxi contains the four typical heating patterns defined in this study. There are differences and commonalities with the heating pattern of the AAHP in Shandong and Shanxi provinces. The common denominator is that both in Shandong and Shanxi, when the living room is not occupied, the occupants will turn off the AAHP, and they choose to switch off the AAHP while they sleep at night.

According to the residents' responses, the heating demand in the living room was dependent on the outdoor temperature and the living room being occupied. When the occupant is not resting in the living room or leaving the home, they prefer to turn off the AAHP. It is easy to understand that they tend to switch off the AAHP when away from home to save on heating operation costs, but it is worth investigating why AAHPs in Shandong and Shanxi are switched off during the coldest winter evenings. There are some common reasons in Shandong and Shanxi provinces for turning off the AAHP. In rural areas, the quilts are usually thicker and warmer, and they do not need any heating equipment; another reason is that the AAHP is installed in the living room but not in the bedroom, resulting in the AAHP being turned off after sleep.

This is not to say that occupants in Shandong and Shanxi provinces do not have a heating demand in their bedrooms. Hence, when the AAHP in the living room is turned on, occupants in these two regions will adopt the same strategy. Bedroom heating is primarily accomplished by opening the bedroom door to guarantee a pleasant interior temperature in the bedroom; however, this type of heating pattern increases the heating cost and is thus rarely used. Because the bedroom is a private space, the door will be closed after sleep. During the early and late heating periods, the outdoor temperature is relatively high, and the indoor temperature is not so cold that it cannot be tolerated; however, as the mid heating period approaches, the surveys showed that air conditioners, electric blankets, and oil-filled electric heaters in the bedroom were frequently used for supplementary heating during the time when people were waking up and before bedtime, as shown in Figure 3. Furthermore, due to the complex rural household structure and large population, the residents in rooms without the AAHP will use this auxiliary heating equipment for heating purposes. Air conditioners, electric blankets, and oil-filled electric heaters for local heating will have an impact on the AAHP's heating pattern, and rural residents' behavior of using space heating equipment (e.g., AAHP) to heat the entire indoor space might be affected by the tradeoff between thermal need and operating costs.

The differences are that the AAHP in Shanxi's living rooms is not only used intermittently but also has a continuous heating pattern that heats 24 h a day. One reason for

this is that, based on their living habits, they were willing to sleep in the living room, and thus the living room serves as a bedroom. For this section of residents, the living room not only becomes a place of daily entertainment, but also an important place to rest and sleep. Another reason is that households in Shanxi have infants in the family, and need to maintain higher indoor temperatures in winter to make them more comfortable; thus, the heating pattern tends to be one of continuous heating for about 24 h.

## 5. Limitations and Future Recommendation

Although the AAHP heating pattern was identified and the characteristics in different heating periods were compared and analyzed, there are still several limitations. In this study, we do not have access to AAHP heating electric power data, which could be used to better analyze heating patterns. Due to the limitation of data access, we chose the indoor temperature parameter as the data to identify rural residents' heating patterns using the AAHP. Another limitation is that we did not collect questionnaires from residents during the heating season and the residents' heating behavior was not analyzed. In the future, we will try to collect more household background information, such as heating energy consumption and household income, to better understand the phenomena discovered in this study.

The predominant rural resident lifestyle and living habit is one of "part-time, part-space" use of a rural building. The heating patterns identified in this study are not only consistent with this lifestyle but also with the economic level of rural residents. The representative heating patterns of the AAHP reflect actual heating demands in rural areas, which could help policymakers reconsider and better understand whether residents are satisfied with the current clean heating policy, whether to solve existing problems in the clean heating process and ensure the sustainable development of clean heating. Large-scale promotion of "coal-to-electricity" clean heating may result in a winter sharp increase in electricity consumption. In addition, the energy management department can adjust the production and scheduling of electrical energy based on the heating demand characteristics of the residents, such as heating period and heating duration, to achieve a dynamic balance between the energy supply side and the consumption side, and to ensure a sufficient energy supply and efficient utilization.

## 6. Conclusions

The purpose of this study was to identify the heating patterns of AAHPs and understand the actual heating demand of rural residents. We classified the whole heating season into three phases and used a clustering analysis to identify the AAHP heating patterns of rural residents in different heating periods. The k-means clustering analysis was useful for identifying typical heating patterns, which aided in understanding the underlying information in the collected data. To solve these issues, a field investigation and long-term indoor environment parameter monitoring were carried out in different rural areas of Shandong, Hebei, and Shanxi provinces, China.

- During three different heating periods, four typical heating patterns of the AAHP were identified through clustering analysis, including (1) Intermittent heating for multiple periods; (2) Intermittent heating for a single period; (3) Continuous heating for 24 h; and (4) Continuous heating for multiple periods.
- Occupant heating behaviors were dominated by outdoor temperature fluctuation. During the mid heating period, the heating operation time periods and heating duration of AAHP were longer than those in the early and late heating periods. The actual heating demand of rural residents is influenced by the outside temperature.
- In the three heating periods, the heating demand is influenced by room occupancy. There was a Chinese holiday during the mid heating period, and families with elderly members and children had a higher heating demand. However, residents' income and awareness of clean heating can all influence their behavior and demand.

- The AAHP installed in the living room is mainly used with an intermittent heating pattern, and occupants will turn them off before going to bed. Half of the AAHPs in residents' bedrooms were continuously heated for 24 h.

**Author Contributions:** Conceptualization, X.C.; Methodology, X.C., Z.L. and M.L.; Investigation, L.D. and W.Z.; Data curation, L.D. and W.Z.; Writing—original draft, X.C.; Writing—review & editing, Z.L. and M.L.; Supervision, M.L.; Project administration, M.L. All authors have read and agreed to the published version of the manuscript.

**Funding:** This research was funded by the National Key R&D Program of China: 2018YFD1100704.

**Data Availability Statement:** Not applicable.

**Conflicts of Interest:** The authors declare no conflict of interest.

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
