# Peer review of "Occupant Heating Patterns of Low-Temperature Air-to-Air Heat Pumps in Rural Areas during Different Heating Periods"

_buildings, doi:10.3390/buildings13030679_

Round 1

Reviewer 1 Report

1)As stated by the paper,  there is a lack of surveys of energy consumption and energy costs,  which are major factors  influencing the way heating equipment is used.

2)In this paper, the survey data of the three regions are analyzed respectively, but the common points and differences reflected in different regions are not discussed. If the region is not the main influencing factor of the behavior pattern, I suggest sorting out the contents of the three regions to make the article more concise.

3)Air conditioners, electric blankets and oil-filled electric heaters are also widely used for local heating in rural area. It is also the influencing factors of the heating behavior pattern. Considering the use and proportion of these heating equipment in the behavior survey will make the results of the behavior survey closer to the actual situation.

Author Response

Dear Editor:

Please see the attachment, Thanks.

Author Response

Dear Editor:

Please see the attachment. Thanks.
